# Pathogenesis-Targeted Preventive Strategies for Multidrug Resistant Ventilator-Associated Pneumonia: A Narrative Review

**DOI:** 10.3390/microorganisms8060821

**Published:** 2020-05-30

**Authors:** Antonella Cotoia, Savino Spadaro, Guido Gambetti, Despoina Koulenti, Gilda Cinnella

**Affiliations:** 1Department of Anesthesia and Intensive Care, University of Foggia, Azienda Ospedaliero-Universitaria Ospedali Riuniti, Viale Pinto 241, 71122 Foggia, Italy; gambetti.guido@gmail.com (G.G.); gilda.cinnella@unifg.it (G.C.); 2Department of Morphology, Surgery and Experimental Medicine, Anesthesia and Intensive Care Section, University of Ferrara, Azienda Ospedaliera- Universitaria Sant’Anna, Via Aldo Moro 8, 44124 Ferrara, Italy; savinospadaro@gmail.com; 32nd Critical Care Department, Attikon University Hospital, 12462 Athens, Greece; d.koulenti@uq.edu.au; 4UQCCR, Faculty of Medicine, The University of Queensland, Brisbane QLD 4029, Australia

**Keywords:** ventilator-associated pneumonia, multidrug resistant, prevention, strategies, intensive care

## Abstract

Ventilator-associated pneumonia (VAP) is the most common hospital-acquired infection in the intensive care unit (ICU), accounting for relevant morbidity and mortality among critically ill patients, especially when caused by multidrug resistant (MDR) organisms. The rising problem of MDR etiologies, which has led to a reduction in treatment options, have increased clinician’s attention to the employment of effective prevention strategies. In this narrative review we summarized the evidence resulting from 27 original articles that were identified through a systematic database search of the last 15 years, focusing on several pathogenesis-targeted strategies which could help preventing MDR-VAP. Oral hygiene with Chlorhexidine (CHX), CHX body washing, selective oral decontamination (SOD) and/or digestive decontamination (SDD), multiple decontamination regimens, probiotics, subglottic secretions drainage (SSD), special cuff material and shape, silver-coated endotracheal tubes (ETTs), universal use of gloves and contact isolation, alcohol-based hand gel, vaporized hydrogen peroxide, and bundles of care have been addressed. The most convincing evidence came from interventions directly addressed against the key factors of MDR-VAP pathogenesis, especially when they are jointly implemented into bundles. Further research, however, is warranted to identify the most effective combination.

## 1. Introduction

Ventilator-associated pneumonia (VAP) is a nosocomial infection of the pulmonary parenchyma which develops in intensive care unit (ICU) patients who have been mechanically ventilated (MV) for at least 48 h [1]. The pathogenesis of VAP is related to several bacterial strains colonizing the oropharyngeal and gastrointestinal tract that reach the lower respiratory tract primarily through the microaspiration of bacterial-laden secretions. Depending on their virulence and the host’s response, they can consequently cause lung infection. Of note, microaspirations are frequent events among critically ill patients and the presence of an endotracheal tube can contribute to lung infection development [2]. Although its estimated incidence (2–16 cases per 1000 ventilator-days) seems to be on the decline [3], probably due to implementation of preventive strategies, VAP is still an important cause of morbidity and mortality among ICU patients. Etiological agents vary from common organisms to multidrug resistant (MDR) pathogens that are difficult to treat. A prospective study found a prevalence of 23.7%, with early-onset VAP commonly associated with *Enterobacteriaceae*, *Staphylococcus aureus*, and late-onset VAP more frequently related to *Pseudomonas aeruginosa*, *Klebsiella pneumoniae*, and *Escherichia coli*. Up to 27.8% had a polymicrobial infection, and 29.2% of the isolated pathogens were MDR [4]. MDR organisms (MDROs) incidence, however, can be even higher as reported by a recent cross-sectional study showing a prevalence of 35% over a two-year period, with 60.87% of the cases caused by MDR pathogens, especially among the late-onset VAP. Independent risk factors for VAP by MDR pathogens were prior antibiotic therapy and hospitalization of five days or more [5]. However, in other studies, etiology between early- and late-onset VAP have been found to be similar [6], emphasizing the role of local ICU ecology as the most important risk factor for acquiring MDR pathogens, irrespective of the length of intubation. Of note, when VAP is caused by MDR microorganisms, it has higher in-hospital and 30-day mortality rates [7,8,9], making prevention strategies and adherence to prevention and treatment protocols even more pivotal (Appendix A) [1,10,11,12,13]. The rising problem of MDR etiologies has, at the same time, led to a reduction in treatment options, shifting the focus on the search for new and effective VAP preventive strategies, which could directly hit its pathogenesis.

Our narrative review, resulting from a systematic database search across 15 years, aims to identify and summarize the most important pathogenesis-targeted strategies to prevent MDR-VAP.

## 2. Materials and Methods

PubMed/MEDLINE, Cumulative Index to Nursing and Allied Health Literature (CINAHL), and Cochrane Library databases were searched for English language full-text articles published between 1th January 2005 and 31th December 2019 with following query: *(((ventilator associated) AND pneumonia) OR VAP) AND (multidrug resistant OR MDR) AND prevention*.

Duplicate references were removed. For the selection phase, prespecified inclusion and exclusion criteria have been defined as follows: original articles providing data regarding any established and/or potential nonantibiotic prevention strategy for MDR-VAP were only included. Nonoriginal articles including previous reviews, case reports, case series, editorials, nonhuman studies and study protocols were excluded. Studies considered not relevant for the aim of our review (i.e., lack of outcomes of interest) were also excluded.

Three authors independently screened titles and abstracts of each citation according the above-mentioned criteria, selecting those for full-text review. Each retrieved full-text article was then independently evaluated for inclusion. The initial search led us to collect 225 articles from which 47 duplicates were excluded. Of the remaining 178, 128 were further excluded (Appendix A). The full text of the remaining 50 articles was reviewed and 23 articles were considered nonrelevant. Therefore 27 articles were considered for the drafting of this review.

For the aim of our narrative review, we extracted and critically summarized all data regarding VAP incidence and MDR colonization rates when available. Secondary outcomes (i.e., antibiotic use, emergence of new patterns of resistance, hospital and ICU length-of stay, costs) have been reported as well, when considered to be of interest (Appendix A).

## 3. Patient Hygiene

### 3.1. Oral Hygiene with Chlorhexidine 

Oral microbiota in ICU patients is different from healthy subjects. In particular, after 48 h of hospitalization, a major shift in its composition takes place: normal Gram-positive bacilli, usually predominant in the oral flora, become substituted by Gram-negative bacilli commonly associated with VAP [14]. Their entry into the lower respiratory tract is then facilitated by aspiration of oropharyngeal and nasopharyngeal secretions around the endotracheal tube cuff, with the dental plaque acting as a constant pathogen reservoir [15]. For this reason, the maintenance of oral hygiene in patients undergoing mechanical ventilation is of the primary importance for the prevention of VAP.

In this context, Chlorhexidine (CHX), the most commonly used product for oral hygiene in mechanically ventilated patients, has proven its effectiveness in reducing VAP rates. A randomized controlled trial (RCT) published in 2017 by Tuon et al. showed how oral mucosa and dental plaque of critically ill patients are rapidly colonized by MDR microorganisms [16]. They also proved how using 2% Chlorhexidine digluconate (CHX) for mouth-rinsing the oral cavity could lead to a significant reduction in the total number of MDR bacteria and a lower percentage of MRSA, compared to the placebo group, both in the oral mucosa and dental plaque cultures. The determination of CHX minimum inhibitory concentration (MIC) for *S. aureus* and Gram-negative bacilli confirmed the high susceptibility of these species for this antimicrobial agent, even after brief exposure (1 h) and at relatively low concentrations [10]. Although this study demonstrated how dental plaque and oral mucosa are rapidly colonized with MDR bacteria and that 2% CHX is able to reduce the incidence of oral colonization by methicillin-resistant *S. aureus* (MRSA), a known potential etiology for MDR-VAP, the small sample size did not allow a full assessment of the influence of CHX on the incidence of MDR-VAP [10]. Thus, further studies should be taken into account to assess the potential role of oral hygiene with 2% CHX in reducing the rate of MDR-VAP. It is important to note, however, that concerns regarding 2% CHX mouthwash have been raised. In 2016, a multicenter cluster-randomized study found an unexpected incidence of oral mucosal lesions in patients receiving 2% CHX (9.8% in the first two hospitals testing this intervention), such that the safety committee recommended to replace 2% CHX with 1% CHX oral gel in the remaining participating centers [17]. When lower CHX concentrations were tested (0.20% and 0.12%), no evidence of oral lesions was found, although we have no data on VAP and MDR colonization rates. However, another retrospective hospital-wide cohort study found 0.05% and 0.12% CHX to be associated with a significant increased risk of death (AUC 0.94), with a surprisingly high number needed to harm of 47.15 (95% CI 45.19–49.11) [18]. This finding was consistent even after stratifying mortality rates for baseline mortality risk, especially among patients with the lowest baseline risk of death. However, no significant association between CHX oral care and mortality was found in cardiothoracic and vascular surgery patients, nor among patients receiving mechanical ventilation [18]. This evidence was corroborated by two recent meta-analyses whose results suggested to reconsider the indiscriminate use of this practice among unselected populations of critically ill patients [19,20]. Authors raising doubts about CHX’s efficacy—both in terms of oropharyngeal decontamination and VAP reduction—and related concerns regarding changes in pathogen’s susceptibility and potential patient harm [21,22], take then a cautious and supportable stand.

### 3.2. Chlorhexidine Bathing & Cleansing

The skin of patients is a well-known reservoir for pathogens associated with hospital-acquired infections, including MDR-VAP. Skin colonization by MDR bacteria can arise even prior to ICU admission (e.g., in the emergency department or from previous medical care) and pathogen migration to the oral cavity and subsequently to the lower respiratory tract may occur during intubation or aspiration of the airway or the oral cavity. Chlorhexidine (CHX) is a broad-spectrum antimicrobial agent, which, unlike many other antiseptics, has residual bacterial activity that may decrease the bacterial burden on patients’ skin and prevent secondary environmental contamination. For its ability in limiting the source, CHX bathing could theoretically decrease the risk of MDR-VAP in mechanically ventilated patients and has been incorporated into some expert guidelines. Even if a reduction in MDR organisms and in hospital-acquired bloodstream infections have been observed by an RCT published in 2013, these results were not replicated and the effect of CHX bathing on other infections remained unclear [23].

A recent cluster randomized, crossover clinical trial by Noto et al. showed how once-daily bathing of critically ill patients with 2% CHX-impregnated cloths could not lead to a significant difference in individual infections compared to nonantimicrobial cloths (adjusted risk ratio in treatment group: 0.94; 95% CI 0.65–1.37; *p* = 0.83) [24]. Moreover, the results of three different post-hoc analysis warned how CHX bathing could even result in a statistically significant increase in possible or probable VAP. However, not only did the rate of clinical cultures positive for MDR organisms not differ between the two treatment allocations, but the small number of VAP events (eight in the control group and 17 in the CHX group) compared to the total number of patients enrolled may have led the authors avoid to explore for any infection-specific difference in the positivity of clinical cultures for MDR pathogens between the two bathing periods. In a similar RCT, Boonyasiri et al. showed no significant difference both in the number or favorable events (defined as persistent negativity of swab samples for MDR microorganisms throughout ICU admission or the negativization of the ones initially positive for MDR pathogens) and in the incidence rates of VAP (6.5 vs. 6.1 episodes per 1000 ventilator-days, *p* = 0.69) with the use of 2% impregnated washcloths [25]. Interestingly, the authors also performed a comparative etiological analysis for each type of infection. In particular, VAP were caused by *Escherichia coli*, *Klebsiella pneumoniae*, *Pseudomonas aeruginosa*, *Acinetobacter baumannii*, *Stenotrophomonas maltophilia*, and *Candida albicans* (all germs with a high likelihood to be MDR) with no significant difference between CHX and nonantimicrobial soap arms. Taken together, the findings of these studies do not seem to support daily cleansing of patients admitted in ICU with 2% CHX in order to reduce healthcare-associated infections, MDR-VAP included.

The evidence seems conflicting even among secondary sources. In 2015, a meta-analysis of six studies suggested how body washing could decrease VAP risk among critical ill patients (RR: 0.73, 95% CI: 0.57–0.93). However, this effect was more prominent in the four before–after studies included, while no statistically significant decrease in VAP incidence was found by the remaining two RCTs [26]. A subsequent meta-analysis by Frost et al. found similar results: although daily bathing with CHX seemed to reduce VAP by approximately 18% (Bayesian RE-IRR = 0.82, 95% CrI 0.57, 1.25), summary estimates for specific study designs were beneficial only for before–after studies and not for RCTs [27]. To better clarify the issue with improved quality of evidence, the same group performed a trial sequential meta-analysis in 2018, which showed how CHX body washing could effectively reduce MDROs burden in the ICU by approximately 18% (DL-RE IRR = 0.82, 95% CI 0.69, 0.98), but had no effect in decreasing VAP rates (DL-RE IRR = 1.33, 95% CI 0.81, 2.18) [28].

Since a state of clinical equipoise is still met, further well-designed experimental studies—even in broader groups of ICU patients thus magnifying external validity—should be encouraged.

## 4. Targeting the Mouth, the Nose, the Stomach, and the Gut

### 4.1. Selective Oral and/or Digestive Decontamination

Selective oral decontamination (SOD) and selective digestive decontamination (SDD) are prophylactic treatments administered in critically ill patients with the aim to prevent or even eradicate potentially pathogenic microorganisms in the digestive tract flora. The unaffected anaerobic bacterial species remaining would further prevent new colonization with potential pathogens. SOD and SDD have been proven to prevent serval infection and reduce mortality in ICU patients; however, it has led to concerns regarding possible selection of antibiotic-resistant strains.

In 2018, Pèrez-Granda et al. published a before–after study in which they examined the impact of SDD for preventing VAP in heart-surgery ICU patients [29]. They found not only a statistically significant decrease in VAP rate after the introduction of SDD, both in the overall study population (16.26/1000 days and 6.80 episodes/1000 days of MV, *p* = 0.01) and high-risk patients (25.85/1000 days and 12.06 episodes/1000 days of MV, *p* = 0.04), but VAP also occurred significantly later among those receiving SDD (6 (3–11) vs. 5 (3–3–14.5); *p* = 0.01). Moreover, SDD did not promote the emergence of antibiotic-resistant microorganisms. On the contrary, it was able to reduce the number of isolates positive MDR or XDR *P. aeruginosa* (*p* < 0.001) while the total number of MDR pathogens did not increase during the study period. Sánchez-Ramírez et al. published another prospective study focused on the efficacy of long-term SDD on a mixed ICU population in a high resistance setting [30]. Notably they found a significant reduction (*p* < 0.001) in the rate of infections caused by MDR bacteria, including VAP (RR 0.43; 95% CI 0.32–0.59), with an associated decrease of antimicrobials consumption. Despite the results showed an increase in the rate of tobramycin- and colistin-resistant colonization by 1000 days, adjusting it to the rate of resistance at the time of ICU admission resulted in a nonsignificant increase in ICU colonization resistance. These studies show how SDD could represent an effective preventive strategy for reducing both the burden of MDROs and the incidence of VAP.

Compared to SDD, SOD alone may have a less certain effect on mortality among critically ill patients, as pointed out by a meta-analysis in 2014, but both have been found to be superior to oropharyngeal CHX, which, in turn, has been linked to an increased mortality [20]. On the other hand, a later meta-analysis by Zaho et al. showed how SOD and SDD had similar effects on mortality, length of hospital and ICU stay, and duration of mechanical ventilation, but SOD led to higher ICU-acquired bacteremia and carriage of antibiotic-resistant organisms [31]. The beneficial effect of SOD and SDD on survival was further confirmed by the most recent meta-analysis of Plantinga et al., with SDD found to be more effective than SOD [32].

Considering these data, ERS/ESICM/ESCMID/ALAT guidelines provide a moderate strength recommendation over the use of SOD and/or SDD, pointing out how SOD is suggested in settings of low antibiotic resistance and consumption. They finally make no formal recommendation over the use of CHX due to the unclear balance between potential reduction in VAP rates but possible increase in mortality [1].

### 4.2. Multiple Decontamination Regimens

The usefulness of concurrent implementation of multiple decontamination protocols has also been assessed. The promising results of an RCT conducted by Camus et al. in 2005 highlighted a reduction of ICU-acquired infections (pneumonia in particular) by the combination of oropharyngeal and gastric decontamination with polymyxin/tobramycin plus nasal muciprocin and chlorhexidine body wash [33]. More recently, through a larger prospective study, the same research group confirmed how the systematic implementation of such protocol could decrease the incidence of all-cause ICU-acquired infections (ICU-AI) [34]. Remarkably among these, the authors registered a three-fold decline in total intubation-related pneumonias, with a statistically significant decrease in *P. aeruginosa* related infections (1.7/1000 vs. 3.5/1000). Acquired infections caused by Enterobacteriaceae resistant to Piperacillin/tazobactam, Ceftazidime, Ciprofloxacin, or Colistin and those caused by aerobic Gram-negative bacilli resistant to colistin were also significantly diminished, along with a nonsignificant decrease in MRSA infections rates. 

Herein, no definitive conclusions can be made about the possible long-term effect on the emergence of new MDR microorganisms, the association of multiple decontamination regimes seems to represent a promising and effective tool for preventing MDR-VAP.

### 4.3. Probiotic Preparations

Probiotics are commercially available living microbial agents, which, when ingested in adequate amounts, have beneficial effects on the health of the host. These effects are exerted by preventing pathogenic bacteria overgrowth into the digestive tract and reducing intestinal hyperpermeability through an improvement in the gut mucosal barrier and a reduction in bacterial translocation. The modulation of local and general immunity through toll-like receptor upregulation also seems to play a role. Minimizing colonization by more virulent species and optimizing host immune defenses, probiotics administration could then represent a promising nonantibiotic intervention to reduce the incidence of VAP, even caused by MDR pathogens. 

In a recent prospective double-blind randomized controlled trial, Mahmoodpoor et al. showed how probiotics could significantly decrease the incidence of microbiologically confirmed VAP, gastric residuals, and duration of hospital and ICU stay, and compared to placebo [35]. However, no statistically significant difference between the two groups was found in gastric and oropharyngeal colonization (29.2% vs. 37%, *p* = 0.26 and 48.5% vs. 63%, *p* = 0.11 respectively), in the incidence of MDR pathogens (12.5% vs. 18.5%, *p* = 0.29), and in Kaplan–Meier survival curves for time to the first VAP episode (log-rank test = 1.89; *p* = 0.17), even if the numbers seemed to be in favor of the probiotic use [35]. Overall, these results showed how probiotic administration may delay VAP occurrence in ICU patients. However, maybe because the majority of patients enrolled in this study were surgical cases and therefore had a short duration of mechanical ventilation, the decrease of VAP occurrence between the two groups was not significant. In addition, VAP cases were identified only with a microbiologically confirmed diagnosis, which could by itself lower the estimated VAP incidence and hence contribute to the negative results. In this trial, the probiotics were able to decrease oropharyngeal and gastric colonization, along with MDR organisms’ incidence during the study, even if this result was not statistically significant. Thus, MDR pathogens appear particularly difficult to eradicate by probiotics alone. Finally, by reducing the gastric residual volume, probiotics could decrease the value of gastric residual monitoring in VAP occurrence, supporting the most recent findings. However, since VAP seems more closely associated with aspiration of contaminated oropharyngeal secretions than of gastric contents, reducing the burden of MDR germs in the first may represent a potential strategy to decrease MDR-VAP. A Cochrane analysis of eight RCTs (*n* = 1083) showed how probiotics could decrease VAP incidence (OR 0.70, 95% CI 0.52–0.95); however, overall quality of evidence was low and the aggregated results for mortality were uncertain [36]. Another more recent meta-analysis by Wang et al., being the largest and most updated evaluation of the overall effects of probiotics for VAP prevention in critically ill patients to date, had similar findings, showing how this strategy was effective in reducing VAP incidence (RR = 0.73, 95% CI = 0.60–0.89; *p* = 0.002) with no significant decrease in mortality, length of stay, and duration of mechanical ventilation [37].

While the overall effect of probiotic preparations seems to be beneficial in terms of VAP rate reduction, the quality of evidence appears relatively low. Future multicenter trials with larger sample sizes and/or implementing probiotics with different strain compositions may be needed to further clarify the usefulness of this nonantibiotic intervention, a field where all possible resources must be sought, considering the increasing incidence of antibacterial resistance and the complex pathogenesis of VAP itself.

## 5. Protecting the Airway

### Silver-Coated Endotracheal Tube

Silver-coated endotracheal tubes (ETT) have been recently proposed to reduce the burden of bacterial airway colonization in mechanically ventilated patients.

In the NASCENT trial [38], the authors demonstrated a statistically significant reduction of VAP in patients intubated for ≥24 h with a silver-coated ETT compared to those receiving a similar but uncoated tube (4.8% vs. 7.5% microbiologically confirmed VAP respectively, with a number needed to treat approximately of 37). Moreover, a delayed time to VAP occurrence and a reduced incidence of VAP caused by potentially highly resistant pathogens (including methicillin-resistant *Staphylococcus aureus*, *Pseudomonas aeruginosa*, *Acinetobacter baumannii*, *Stenotrophomonas maltophilia*, and *Burkholderia cepacia*) have also been registered in the intervention group, although with no statistically difference in the duration of intubation, length of hospital or ICU stay, and mortality. A cost-effectiveness analysis of silver-coated ETT showed a savings of US$12,800 per case of VAP prevented [39]. To further explore the clinical effect of the silver-coated ETT, Afessa et al. performed a retrospective cohort analysis in patients who developed VAP in the NASCENT study. They also focused the attention on the causative etiologies, particularly potentially MDR pathogens [40]. The authors found the silver-coated ETT to be associated with a statistically significant reduction in mortality in patients diagnosed with VAP (14% vs. 36%, *p* = 0.03) and a lower incidence of potentially MDR germs as the cause of death in this subgroup. Moreover, considering all patients enrolled in the NASCENT study, the authors found a 50% lower rate of VAP caused by potentially MDR bacteria in the silver-coated ETT group than in those with conventional ETT.

Collectively, these studies seem to suggest that the observed association between the silver-coated ETT and decreased mortality in patient with VAP could be related to reduced bacterial burden, particularly from potentially MDR pathogens. This hypothesis certainly deserves further evaluation.

## 6. Targeting Transmission

### 6.1. Universal Gloving and Contact Isolation

Contact precautions have been recommended in view of the emergence of MDR organisms such as vancomycin-resistant *Enterococci* (VRE) and MRSA; however, barriers to widespread implementation exist and the optimal strategy remains controversial. For this reason, prevention techniques that are easier to apply such as the universal use of nonsterile gloves, have been proposed. 

In a controlled trial, Bearman et al. (2007) showed how gloving could guarantee a higher compliance among the operators compared to comprehensive contact precautions (gown and gloves), despite a lower compliance for hand hygiene [41]. However, the authors registered an increased incidence in nosocomial infection rates during the universal gloving phase of the study, VAP included (0 vs. 2.3 episodes per 1000 device days, *p* < 0.001). Moreover, no differences in MRSA and VRE colonization were observed between the two study periods. 

After aggressive hospital-wide hand hygiene education, three years later, the same group published another before–after trial using emollient-impregnated gloves for the universal gloving phase, which this time led to higher hand hygiene compliance [42]. Again, they found no differences in MRSA and VRE acquisition between the two study periods, but no discrepancy in device-associated infection was found (for VAP: 1.0 vs. 1.1 cases/1000 device days, *p* = 0.09).

Summarizing the results of these studies seems to highlight the need of measures to increase hand hygiene compliance in order to minimize the risk of nosocomial pathogens transmission, before letting gloving (a substantially less burdensome strategy) replace contact precautions.

Against the latter, patient safety concerns have also been raised, based on the evidence that patients placed in contact isolation are exposed to a higher risk of medical errors and adverse events, as demonstrated in a study published in 2013 by Zahar et al. [43]. Notably, among these adverse events, the authors reported a significant increased incidence of MDR-VAP in isolated patients (sHR 2.1, 95% CI 1.3–3.3, *p* = 0.002). 

In this field, the optimal preventive strategy seems so far to be defined. In addition, the identification and correction of nonadherent practices should in any case become a standard component of infection prevention and control bundles.

Similar conclusions can be drawn when looking to the most recent meta-analysis, showing how universal gloving may provide a small protective effect on healthcare-associated infections in pediatric/neonatal ICU patients (IRR 0.75; 95% CI, 0.65–0.87), but not in adult ICUs (IRR 1.01; 95% CI 0.91–1.13) [44]. However, this difference may also have been due to different care patterns or increased awareness among healthcare workers when dealing with the pediatric population, and it was not observed in the pooled analysis of RCTs only [44]. Moreover, another meta-analysis evaluating discontinuation of contact precautions for MDROs did not find increased infection rates by MRSA and VRE, with discordant results for ESBL and a trend toward an increase in *C. difficile* infections but with a very low rate of transmission [45]. De Angelis et al. had similar findings in 2014, when their meta-analysis showed how contact precautions were unable to significantly reduce VRE acquisition rate (pooled RR 1.08, 95% CI 0.63–1.83, I2 0%), compared to hand hygiene (pooled RR 0.53, 95% CI 0.39–0.73, I2 26%) [46]. In the same year, a Chinese group published a further meta-analysis highlighting the beneficial role of enhancing hand hygiene to prevent VAP (pooled OR = 2.23, 95% CI 1.62–3.07, *p* < 0.00001) [47], fully justifying Spanish guidelines that presented it as a basic mandatory strategy just one year before [11].

### 6.2. Alcohol-Based Hand Gel

Despite the fact that appropriate hand hygiene is generally accepted as a cardinal measure to prevent healthcare-associated infections, ICU personnel’s adherence to hand hygiene is often suboptimal. In the attempt to increase hand hygiene compliance, effective and less time-consuming products have been approved, such as alcohol-based hand gel. However, there is a paucity of data regarding its effectiveness in changing outcomes of critically ill patients.

In a cross-over RCT of 2008, Rupp et al. showed how the use of alcohol-based hand gel resulted in a significant and sustained improvement in the rate of hand hygiene adherence, but not resulting in a significant relationship with the rates of VAP or infection due to *MRSA*, *VRE*, or *C. difficile* [48]. Moreover, the use of rings, fingernails length > 2 mm and lack of access to hand gel were all associated with increased microbial carriage on hands.

The same authors warned the reader not to interpret these results as a proof of hand hygiene inefficacy, but rather as a signal that trying to decrease MDR infection rates just through the use of alcohol-based hand gel is, in the end, simply unrealistic. 

## 7. Targeting the Environment

### Environmental Decontamination with Vaporized Hydrogen Peroxide

Vaporized hydrogen peroxide (VHP) represents a new approach for environmental disinfection that was used initially to interrupt outbreaks caused by *MRSA*.

Ray et al. combined the use of VHP with infection control measures to break the cycle of nosocomial transmission of MDR *A. baumannii* in a long-term acute care hospital (LTACH) in 2008 [49]. In their observational study, the authors found how VAP was the most common clinical syndrome caused by the MDR *A. baumannii* during the outbreak, with an associated mortality of 14%. 

Despite the small number of cases and the nontraditional setting, we believe that VHP decontamination could represent a useful strategy to tackle MDR germ outbreaks even in ICU, and should be considered as an add-on to more conventional preventive measures.

## 8. What Else Guidelines Recommend

### Other Nonpharmacological Interventions

Although not identified by our database search, several other nonpharmacological interventions are recommended by different guidelines and deserve to be mentioned [10,11,12,13].

Since VAP pathogenesis is closely related to invasive ventilation, strategies favoring noninvasive positive pressure ventilation (NIPPV) in selected patients, minimizing sedation, and contemplating the daily assessment for readiness to extubation are recurring themes in many prevention bundles [10,11,12,13]. Whenever extubation is not possible, meticulous handling of other ETT-related risk factors should be undertaken, with strict hand hygiene and education for appropriate airway management being mandatory [11]. 

Guidelines heavily agree on over the head of the bed elevation (semirecumbent position)—avoiding 0° whenever possible—and aspiration of subglottic secretions even with the use of ETTs with appropriate drainage ports [10,11,12,13]. In 2007, Lorente et al. compared the efficacy of an ETT with polyurethane cuff and subglottic secretions drainage ports (ETT-PUC-SSD) against conventional ETT in reducing VAP incidence [50]. Although ETT-PUC-SSD was effective in preventing both early and late-onset VAP, we cannot ascertain if these positive results could be attributed more to the improved suctioning system or to the different cuff material. The effectiveness of this preventive strategy was better supported by a large multicentric RCT of 2010, in which VAP rate was significantly lower among patients intubated with an ETT allowing subglottic secretions drainage (SSD) [51]. These results were corroborated by a further RCT which linked subglottic secretions suctioning to a significant reduction in VAP prevalence and in antibiotic consumption [52]. Several further studies and a very recent meta-analysis led to similar results [53,54,55,56]; however, some failed to show a decrease in clinical and microbiological VAP incidence, maybe due to small sample sizes [57,58,59]. Some authors have also questioned what the proper modality should be, even in terms of timing or system design. A small RCT by Fujimoto et al. suggested that continuous SSD could decrease mechanical ventilation days and ICU stay, compared to intermittent SSD, even if it was not associated to a significant reduction in VAP incidence [60]. Kolobow et al. suggested indeed that a novel design called “mucus slurper” could allow better secretion aspiration, through the use of several holes placed near the ETT tip [61]. To our best knowledge, the literature still provides no clinical data about its effectiveness. Avoiding secretions descent into the lower respiratory tract may also be granted by regularly verifying ETT cuff pressure and is strongly suggested by several guidelines [11,12,13]—even through automated devices [10]. The use of ultrathin polyurethane ETT cuffs is presented as a special approach with low quality of evidence in the 2014 SHEA/IDSA practice recommendations update [10], while SFAR–SLRF 2017 guidelines advise against the use of antiseptic-coated ETTs or tubes with an “optimized” cuff shape [12]. Due to its well-recognized role as a gateway to ventilator-associated pneumonia, several tracheal tube features have in fact been studied for years [62]. While the standard ETT cuff shape is spherical or cylindrical, other designs have been tested in the belief that they could prevent microaspirations through an improved sealing of the trachea. The most studied one has been the tapered cuff shape; however, clinical evidence does not support its use for VAP prevention, as they seem not to be effective in reducing both microaspirations and VAP rates [2,63,64,65]. The two most recent meta-analyses draw the same conclusions [66,67]. Different materials have also been tested. Standard ETT cuff is usually made out of PVC, but alternatives such as silicone, polyurethane, and Lycra have been employed after some encouraging evidence from a preclinical setting [68,69,70,71]. Despite the first clinical studies showed an association between the polyurethane cuff and a lower rate in microaspirations and postoperative pneumonia after cardiac surgery [72,73], more recent trials by the groups of Philippart and Suhas did not show any benefit in terms of VAP occurrence [63,74]. Silicone is a very compliant material that was believed to allow a higher intracuff pressure, limiting the one against the upper airway surface. For this reason, they are often referred as “pressure-limited cuffs”. A large RCT by Gopal et al. showed a significant reduction in VAP incidence among cardiac surgery patients [75]. A secondary analysis suggested how this effect could have been mostly due to a reduction in late-onset VAP, as the benefit in ETT colonization was more evident among patients undergoing prolonged mechanical ventilation [76]. Finally, no-pressure sealing systems have also been described, in line with the “ultrathin-walled” ETT designed by Reali-Forster et al. which was originally equipped with 12–20 polyurethane gills close to the tube tip, in place of the conventional and inflatable cuff [77,78]. However, to the best of our knowledge, their role in VAP prevention has not yet been proved in the clinical setting. Not less important, scheduled ventilator circuit changes are universally discouraged and should be performed only when visible soiling or malfunctions are detected [10,11,12,13]. Finally, early tracheostomy (<7 days) is generally not recommended for VAP prevention, but may be indicated for different rationales [10,12]. Whenever possible, physiotherapy and early mobilization should be encouraged [10].

To address risk factors related to oropharyngeal and digestive tract colonization—another key component in VAP pathogenesis—the guidelines suggest several interventions. While several interventions have been mentioned in the previous sections, other nonpharmacological preventive strategies should be taken into account as well. Apart from specific indications, guidelines do not recommend the use of stress ulcer prophylaxis [10,12], routine monitoring of gastric residuals, and parenteral nutrition with the aim to prevent VAP [10]. The SFAR–SLRF 2017 guidelines encourage initiating enteral feeding within the first 48 h of admission, avoiding postpyloric feeding, except in selected cases [12].

## 9. Combining the Strategies

### Bundles of Care

Considering the complex pathogenesis of MDR-VAP and its multiple risk factors, the implementation of structured prevention bundles could confer a greater benefit than the employment of single and disjunct preventive strategies. Both the individual susceptibility of the mechanically ventilated patient and environmental factors, in fact, need to be targeted at the same time.

During the years 2001–2002, an alarming outbreak of MDR *Acinetobacter baumanni–calcoaceticus* at the Royal London Hospital, was properly controlled by putting into practice a bundle of control measures improving both patient care and environmental hygiene [79]. Combining a ventilator bundle developed in accordance to the Institute of Health Improvement guidelines with an oral care bundle resulted in a significant decrease in ICU length of stay, total ventilator days, and VAP rate (from the average of 8.2 per 1000 ventilator days for 13 months to 3.3 per 1000 ventilator days for 24 months, *p* = 0.02) between 2003 and 2006 [80]. Landrum et al. showed how the implementation of aggressive and wide-ranging infection control interventions, including many recommendations by the American Thoracic Society and Infection Disease Society of America, led to marked reduction in the rate of VAP (from 60.6 to 11.1 per 1000 ventilator days, *p* = 0.029) with a sustained improved in the following two months of targeted surveillance [81]. Moreover, they observed a significant improvement in the antimicrobial susceptibility of MDR *A. baumannii*, the most common pathogen causing VAP in that setting, which the authors ascribed to a reduced antimicrobial selection pressure likely due to the restriction of antibiotics used for surgical prophylaxis, as a part of their infection control plan. Walkey et al. demonstrated between 2006 and 2009 that a bundle of interventions based on Centers for Disease Control and Prevention (CDC) guidelines could be effective in preventing VAP, which, in 95% of the cases, was caused by MDR germs, even in the setting of a long-term acute care hospital [82]. As a matter of fact, they registered a 56% reduction in the rate of VAP (from 3.8 to 1.67 cases per 1000 ventilator-days after the implementation of the VAP-bundle approach, *p* < 0.001). In another prospective study, an infection control education program in a tertiary care hospital in Pakistan resulted in a nonsignificant decrease in VAP rates (18% to 13%, *p* = 0.11) [83]. This may highlight the need for established infrastructure for effective infection control, which may be lacking in a hospital of a developing country (as it is even proved by the fact that two thirds of VAP were polymicrobial, with an high incidence of MDR germs). While the implementation of preventive infection control bundles in another middle-income country hospital was associated with a nonsignificant decrease in VAP rates after a eight-year period (incidence rate ratio, IRR = 0.88, *p* = 0.574), it led to a significant reduction in *MRSA*, *P. aeruginosa* and *A. baumannii* colonization (IRR 0.13, *p* < 0.001 after three years; IRR 0.63, *p* = 0.002 after eight years and IRR 0.53, *p* < 0.001 after eight years, respectively) [84]. Employing a standardized VAP preventive bundle composed of many interventions suggested by the published guidelines, Righi et al. reduced VAP incidence from 15.9% to 6.7% (*p* < 0.001) over a period of seven years, both early-onset VAP (6.6% to 1.9%; *p* < 0.001) and late-onset VAP (9.3 to 4.7%; *p* = 0.001) [85]. Moreover, adding SDD to the bundle significantly decreased the risk of developing MDR-VAP as well (odds ratio, OR 0.54; 95% CI, 0.31–0.91). Between 2012 and 2013, Gao et al. developed a bundle for the control of hospital-acquired infections based on several guidelines which allow a decrease in VAP incidence (from 32.72/1000 to 24.60/1000) and in percentage of MDR pathogens (from 67.91% to 59.68%) [86]. A retrospective study on epidemiology and outcome of VAP in an heterogeneous ICU population in Qatar showed a progressive decrease in VAP incidence rates (5.42 per 1000 ventilator-days in 2010, 5.91 per 1000 ventilator-days in 2011, and 3.88 per 1000 ventilator-days in 2012), which could be attributed to an improved compliance to the VAP prevention bundle strictly employed within that hospital facility [87]. However, being a retrospective study, the adherence rates were not measured and a definite conclusion cannot be drawn. An analysis of the OUTCOMEREA network showed a significant decrease in early-onset VAP over the study time periods, which the group ascribed to changes in prevention measures used in the study centers [88]. However, the lack of monitoring of the individual compliance with these strategies may hide the role of unmeasured confounding factors. Another prospective study by Khurana et al., designed to assess the impact of VAP and MDR germs on critically ill patients, highlighted the inverse correlation between VAP incidence and preventive measures compliance (ventilator bundle plus hand hygiene) trends [89]. Again, the decline in VAP rate observed in young and old patients admitted to French ICUs between 2007 and 2014 (aIRR 95%CI 0.88, 0.82–0.94, *p* < 0.001 and 0.89, 0.81–0.98, *p* = 0.022, respectively) was partially explained by the improvement of ventilator bundle adherence over time [90]. Regarding the MDR etiologies, they detected a significant decrease in MRSA-related VAP incidence in all age groups accompanied by a stable and a reduced CRPA-related VAP incidence in the young group and in the old group, respectively (−47.9%, *p* = 0.002). Moreover, the warned about an increase in 3GCRE-related VAP incidence rates in all age groups, with the highest increase seen in the young group (+96.2%; *p* < 0.001) and the lowest increase in the very old group (+54.3%, *p* = 0.002). Finally, a retrospective study by Kanafani et al. reported a significant reduction in VAP rates after the combined adoption of a ventilator bundle and multiple aggressive infection control strategies (from 13.1 in 2008 to 1.1 per 1000 ventilator-days in 2017, with a reduction rate of 91.6%) [91]. The same measures seemed also capable to decrease MDR *A. baumannii* colonization pressure in ICU patients during an outbreak between 2012 and 2014 (from 361.6 per 1000 patient days to 322.0 per 1000 patient days) in the same hospital [92].

On the complex issue of preventing MDR-VAP, considerable evidence seems in favor of adopting structured prevention bundles matched with other aggressive infection control strategies as needed.

## 10. Discussion

The efficacy of 2% CHX-based hygiene protocols has been studied both for oral decontamination and bathing, with daunting results [16,24,25]. While we feel like encouraging the conduction of further RCTs to assess the potential role of oral hygiene with CHX in reducing MDR-VAP—considering the well-documented etiological role of microaspirations in VAP pathogenesis—this might not be the case for CHX-based washing and cleansing. 

Evidence supporting SOD and/or SDD as effective techniques to prevent both VAP and decrease MDR colonization pressure is indeed clearer for specific settings, i.e., of low antibiotic resistance. In addition, in light of the findings of our review, concerns regarding potential SDD-driven selection of MDR strains should be demystified [29,30]. Similar considerations can be drawn when exploring what happens when multiple decontamination regimes are jointly employed. The pertaining studies of Camus et al. [33,34] may bring an open question: which of those regimens has the major impact in reducing MDR-VAP? According to our review’s results and taking into consideration VAP pathogenesis, it may sound tempting to state that the reduction in MDR-VAP is attributable to oropharyngeal and gastric decontamination to a greater extent than to CHX body washing. A definite answer, however, could only be given by RCTs with more complex study design.

Probiotics have proven their efficacy in reducing hospital and ICU length of stay, gastric residuals, and microbiologically confirmed VAP [35]. A nonsignificant reduction in oropharyngeal and gastric colonization with MDR germs and a delay in VAP presentation have also been observed. A thoughtful interpretation cannot refrain from considering the pathogenetic role of microaspirations and digestive tract colonization with MDROs in MDR-VAP. Even with some limitations, the study of Mahmoodpoor et al. suggests that probiotics may make a useful contribution in MDR-VAP prevention. Larger trials—eventually testing probiotics with different compositions—would be helpful to support this hypothesis, while keeping in mind that overcoming MDR colonization through their use only is likely to be unrealistic. 

Encouraging evidence come also from the studies investigating the clinical effects of silver-coated ETTs. They have proven their efficacy in reducing VAP incidence and mortality in patients diagnosed with VAP, probably through a decrease in MDR bacterial burden [38,40]. Additionally, their advantages in terms of cost-effectiveness should also be taken into consideration [39]. Further studies are welcomed in order to see these effects on a larger scale.

The role of universal gloving, which has been proposed in the wake of patient safety concerns related to contact isolation, is controversial [43]. While universal gloving may be a less burdensome strategy for healthcare operators, it may lead to an increased risk of healthcare-associated infections if not supported by adequate hand hygiene compliance [41]. Any measure to improve the latter, however, such as use of alcohol-based hand gel, may have no power to decrease MDR burden if solely applied.

When MDRO outbreaks occur, often as a cause of VAP, valuable help may come from the use of VHP for environmental decontamination [49]. Infection control teams should encourage its implementation as an add-on to more conventional measures, according to local epidemiology.

## 11. Conclusions

Among the several prevention strategies for MDR-VAP that have been explored in our review, the most promising ones seem to be those more closely addressing VAP pathogenesis and MDR overgrowth. Preventing oropharyngeal tract colonization with MDR strains and their descent into their airway seem to be of key importance for decrease MDR-VAP incidence. The multifaced ecology of the ICU and the exponential interaction between individual and environmental risk factors —especially in patients undergoing MV—should also be considered. As often happens in medicine, when dealing with such complex diseases, little is achieved with simple and discrete interventions. Not surprisingly, the most convincing conclusions can be drawn from the studies exploring the efficacy of implementing multiple strategies into structured bundles. They have thus proven how combining multiple preventive techniques can, in many cases, lower both VAP rates and MDR pathogens colonization pressure, even resulting in a decreased incidence of MDR-VAP [27,28,29,30,31,32,33,34,35,36,37,38,39,40]. The effectiveness of such bundles should be tested in well-designed trials. Lastly, the appraisal of those studies undertaken in developing countries leads to the reasonable assumption that the presence of established infrastructure for infection control is always essential to maximize their efficacy.

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
