# Peer review of "Pathogenesis-Targeted Preventive Strategies for Multidrug Resistant Ventilator-Associated Pneumonia: A Narrative Review"

_microorganisms, 2020, doi:10.3390/microorganisms8060821_

Round 1
Reviewer 1 Report
This narrative review covers studies from 200x to the present and identifies several themes within the 27 studies reviewed focussing on strategies to prevent MDR-VAP in the ICU.
The review is extensive, detailed and convincing in its presentation of the evidence and the conclusions provided. The review is complete and requires no further revision.
Minor point: page 10, Line 482, Spelling. Qatar, not Quatar.
Reviewer 2 Report
Cotoia et al. reviewed 27 original articles, identified through a systematic database search in the past 15 years. They focused on addressing several pathogenesis-targeted strategies which could prevent patients from MDR-VAP such as oral hygiene with Chlorhexidine (CHX), CHX 25 body washing, selective oral decontamination (SOD) and/or digestive decontamination (SDD), multiple decontamination regimens, probiotics, subglottic secretions drainage (SSD), special cuff material and shape, silver-coated endotracheal tubes (ETTs), universal use of gloves and contact isolation, alcohol-based hand gel, vaporized hydrogen peroxide and bundles of care. The most convincing evidence came from interventions directly addressed against the key factors of MDR-VAP pathogenesis. It seems the best strategy to prevent MDR-VAP inside the complex ICU environment if several effective preventive measures into bundles are jointly implemented. Only the implementation of multiple strategies into structured bundles could maximize benefit. However, the effectiveness of such bundles should be tested in well-designed trials. Below are my comments:
- The section of “abstract” should be summarized and integrated from three to one paragraph.
- In this narrative review, the authors had a “result” section which is divided into 3.1-3.11 subsections. It is not usual for a review article to have the section of “result”. Please divide and arrange the “result” section into several sections and subsections with appropriate titles and subtitles to make the manuscript more reasonable, logical and organized.
- It is somewhat messy, trivial and not organized in the subsection of 3.2, 3.3, 3.5, 3.6, 3.10, and 3.11. Please focus on addressing one topic in one paragraph.
- Please add a section to address and evaluate the roles which in vitro diagnostic medical devices (e.g., electronic nose) can play to prevent MDR-VAP
- Please make a table to compare the advantages and disadvantages (or limitations) of these pathogenesis-targeted strategies you address in the manuscript.
- In the section of discussion, the first two paragraph (line 512-518) should be revised and moved to the section of “introduction”.
- In the section of discussion, the paragraph (line 557-563) should be revised and moved to the section of “conclusion”.
Round 2
Reviewer 2 Report
The manuscript has been significantly improved and I recommended it to be accepted. However, I suggest that Table 3 be placed in an appropriate space in the discussion section, not as a supplementary material.